# Pathobiological Features and Therapeutic Opportunities Linked to TNF Family Member Expression in Classic Hodgkin Lymphoma

**DOI:** 10.3390/cancers16234070

**Published:** 2024-12-05

**Authors:** Mohamed N. Alibrahim, Annunziata Gloghini, Antonino Carbone

**Affiliations:** 1Faculty of Medicine, Zagazig University, Zagazig 44511, Egypt; 2Department of Avanced Pathology, Fondazione IRCCS, Istituto Nazionale dei Tumori, 20133 Milano, Italy; annunziata.gloghini@istitutotumori.mi.it; 3Centro di Riferimento Oncologico, Istituto di Ricovero e Cura a Carattere Scientifico, National Cancer Institute, 33081 Aviano, Italy

**Keywords:** TNF family, CD40, CD30, classic Hodgkin lymphoma, tumor microenvironment, pathobiology, new therapies

## Abstract

The tumor necrosis factor (TNF) family plays a crucial role in the pathogenesis of Hodgkin lymphoma. Hodgkin Reed–Sternberg cells, the diagnostic hallmark of the disease, exhibit overexpression of TNF receptor family members such as CD30 and CD40. These molecules have a critical roles in the survival and proliferation of Hodgkin Reed–Sternberg cells. Therefore, targeting these TNF receptors represents a promising therapeutic strategy. Therapies that target CD30 have already shown efficacy in clinical settings. Checkpoint inhibitors such as nivolumab and pembrolizumab have demonstrated high response rates in patients with classic Hodgkin lymphoma, particularly in those who have not responded to conventional therapies. By understanding how TNF signaling interacts with immune checkpoints, researchers can design more effective treatment regimens that simultaneously target multiple pathways.

## 1. Introduction

This review adopts a narrative approach to synthesize existing evidence on the role of the tumor necrosis factor (TNF) family in the pathogenesis and treatment of Hodgkin lymphoma (HL). The literature search was conducted in November 2024 using major academic databases, including PubMed and Google Scholar. Articles were included based on their relevance to the biological functions of TNF family members, their interactions with Hodgkin Reed–Sternberg cells (HRS), and their therapeutic implications in HL. This narrative review aims to provide a comprehensive discussion of these topics to guide future research and clinical applications.

The TNF family comprises a complex group of cytokines and their receptors that play critical roles in the regulation of immune responses, inflammation, and cell survival. This family includes 19 ligands and 29 receptors, which interact through a complex signaling network that influences various cellular activities, including proliferation, differentiation, and apoptosis [1,2,3,4,5] (Table 1).

One of the most well-studied members of the TNF family is TNF-α, which is known for its pro-inflammatory properties [6,7]; TNF-α can exert both tumor-suppressive and tumor-promoting effects depending on the context, highlighting its dual role in cancer biology, which has led to the exploration of TNF-α as a target for immunotherapeutic strategies [8]. For instance, TNF-α binding to Tumor Necrosis Factor Receptor 1 (TNFR1) typically induces apoptosis in tumor cells and inflammation, while TNFR2 signaling can promote cell survival and proliferation, thus contributing to tumor progression [8,9].

The TNF family also includes several other important members such as BAFF (B-cell-activating factor) and APRIL (a proliferation-inducing ligand), which are crucial for B cell survival and differentiation [10,11]. Dysregulated BAFF expression has been shown in murine models to lead to autoimmune-like conditions, including an SLE-like phenotype, through its impact on B cell homeostasis and autoantibody production [12]. TNF family members are essential for the development and activation of both innate and adaptive immune responses [13]. They facilitate T cell co-stimulation and B cell activation, thereby enhancing the immune response against pathogens [6,10,13]. The signaling pathways activated by TNF receptors, such as nuclear factor-kappa B (NF-κB), are critical for orchestrating these immune processes [4]. Additionally, the TNF family has been implicated in therapeutic strategies for cancer and inflammatory diseases, highlighting their significance in both health and disease [14,15].

The TNF family plays a crucial role in the pathogenesis of HL, particularly through its influence on the TME. HRS cells, the hallmark of cHL, exhibit overexpression of TNF receptor family members such as CD30 and CD40 [16,17,18,19,20], which are integral to their survival and proliferation within the TME. This overexpression is linked to the constitutive activation of NF-κB and alterations in the Janus kinase/signal transducer and activation of transcription (Jak/STAT) signaling pathway, contributing to immune evasion and tumor growth [21,22]. Furthermore, the TME in classic HL is characterized by a diverse group of inflammatory and immune cells, which interact with HRS cells and can modulate responses to therapies, including checkpoint blockade [23]. The unique characteristics of the TME in HL, influenced by TNF family signaling, underscore the complexity of therapeutic strategies aimed at this malignancy [23,24].

Starting from 1985, significant advancements in the understanding of the pathobiology of HL have been achieved [19]. Since 1995, research has focused on the characterization of HL, particularly on the expression of functional CD40 on HRS cells. CD40 was found to play a crucial role in tumor growth and signaling pathways [25]. This work laid the groundwork for understanding the TME in HL. By 2010, the emphasis had shifted towards the complex interactions within the TME, highlighting how immune cells contribute to tumor growth and immune evasion in classic HL (cHL) [21].

The role of cytokines and the JAK-STAT signaling pathway became central to understanding the immune landscape surrounding HL, which was critical for developing targeted therapies.

Subsequently, the focus has expanded to include the implications of checkpoint blockade therapies in HL, revealing how the TME influences treatment outcomes. Studies indicate that the effectiveness of Programmed Cell Death Ligand 1 (PD-1/PD-L1) blockade varies significantly among different lymphomas, underscoring the importance of the TME in therapeutic responses [23,26,27,28].

This evolution illustrates a growing appreciation of the interplay between tumor biology and immune modulation over nearly three decades. The aim of this study is to examine the role of TNF family members in the pathogenesis and treatment of cHL. It focuses on how TNF family signaling, particularly through CD30, CD40, and TNF-α, influences the survival and proliferation of HRS cells within the TME. Additionally, the study explores therapeutic opportunities, including immunotherapies and checkpoint inhibitors, that target these pathways, offering potential advancements in the treatment of cHL.

## 2. Early Research on TNF Family Members in cHL

CD40, a member of the TNF receptor superfamily, is expressed on HRS cells and has been identified as a functional receptor capable of transducing growth signals. Ref. [25] demonstrated that CD40 is expressed in 100% of HL cases examined, highlighting its widespread presence in HRS cells [25]. This expression is significant because it suggests that CD40 may play a pivotal role in the biology of HL, particularly in how these malignant cells evade apoptosis and sustain their proliferation. The interaction of CD40 with its ligand, CD40L, which is expressed on activated T cells, can lead to various downstream signaling cascades that promote cell survival and proliferation, including the activation of NF-κB pathways [25,29].

The survival advantage provided by CD40 signaling in HRS cells is multifaceted. Activation of CD40 can lead to the upregulation of anti-apoptotic proteins, thereby inhibiting programmed cell death. This is particularly relevant in the context of HL, where HRS cells often exhibit resistance to apoptosis, a hallmark of cancer cell survival. Furthermore, CD40 signaling can enhance the production of cytokines and chemokines that create a supportive TME, facilitating the further growth and survival of HRS cells [25].

The ability of CD40 to modulate immune responses also plays a role in the evasion of immune surveillance, allowing HRS cells to persist in the host despite the presence of an immune response.

In addition to CD40, CD30 is another TNF receptor that is characteristically expressed in HRS cells. CD30 signaling has been implicated in promoting cell proliferation and survival through similar mechanisms, including the activation of NF-κB and other survival pathways. The co-expression of CD30 and CD40 in HRS cells suggests a synergistic effect where both receptors contribute to the malignant phenotype of these cells. The interplay between these receptors and their respective ligands can create a complex signaling network that supports the aggressive nature of HL [25].

Given the critical roles of CD30 and CD40 in the survival and proliferation of HRS cells, targeting these TNF receptors represents a promising therapeutic strategy. Inhibiting CD40 signaling could disrupt the survival signals that HRS cells rely on, potentially leading to increased apoptosis and reduced tumor growth. Similarly, therapies that target CD30 have already shown efficacy in clinical settings, as evidenced by the success of brentuximab vedotin, an antibody–drug conjugate targeting CD30, which has demonstrated significant responses in patients with relapsed or refractory HL [25,30].

The therapeutic implications of targeting TNF receptors, particularly CD30 and CD40, may enhance treatment outcomes for patients with HL. The expression of these TNF family receptors plays a crucial role in the biology of HRS cells by contributing to their survival and proliferation, underscoring CD40 as a potential therapeutic target. As research continues to clarify the signaling networks involved in HL, targeting these TNF receptors could offer new treatment strategies aimed at improving patient outcomes [25].

## 3. The Role of the Tumor Microenvironment

The TME plays a pivotal role in the progression and survival of tumors, particularly in cHL, where the interplay between malignant HRS cells and the surrounding non-malignant cells is critical. The TME is composed of various cell types, including immune cells, fibroblasts, and endothelial cells, which collectively influence tumor behavior and immune evasion mechanisms [31]. Understanding the interactions between HRS cells and the TME, especially through the TNF family receptors such as CD30 and CD40, is essential for developing targeted therapies that can modulate these interactions to improve patient outcomes. The CD30 receptor, a member of the TNF receptor superfamily, is expressed on HRS cells and has been shown to mediate various signaling pathways that promote cell survival and proliferation [21]. When CD30 interacts with its ligand, it activates downstream signaling cascades that can inhibit apoptosis and enhance the survival of HRS cells within the TME. This interaction not only supports the malignant cells but also alters the behavior of surrounding immune cells, leading to an immunosuppressive TME that favors tumor growth and immune evasion [21].

The presence of CD30 in the TME is thus a double-edged sword; while it promotes the survival of HRS cells, it also contributes to the recruitment and activation of regulatory T cells and other immune suppressive elements that further protect the tumor from immune surveillance. Similarly, CD40, another TNF family receptor, plays a significant role in the interactions between HRS cells and the immune components of the TME. CD40 is typically expressed on antigen-presenting cells and is crucial for T cell activation and the generation of an effective immune response. However, in the context of cHL, the engagement of CD40 on HRS cells can lead to the secretion of various cytokines that modulate the immune response, often skewing it towards an anti-inflammatory profile. This results in a microenvironment that not only supports tumor growth but also inhibits effective anti-tumor immunity [21].

The ability of HRS cells to manipulate CD40 signaling underscores the complexity of the TME and highlights the potential for therapeutic interventions that target these pathways. The interplay between HRS cells and the TME is further complicated by the presence of various cytokines and chemokines that are produced by both malignant and non-malignant cells. These soluble factors can create a feedback loop that enhances tumor survival and promotes immune evasion. For instance, the secretion of interleukin-10 (IL-10), which has a potent anti-inflammatory effect, by HRS cells can inhibit the function of effector T cells and promote the differentiation of regulatory T cells, thereby dampening the overall immune response against the tumor, Furthermore, IL-10 functions as a growth and differentiation factor for B cells [5,21].

This dynamic interaction between cytokines and the cellular components of the TME is a critical area of research, as it offers insights into potential therapeutic targets that could disrupt these pro-tumorigenic signals. Therapeutic strategies aimed at modulating the TME in cHL are increasingly focused on disrupting the interactions mediated by TNF family receptors like CD30 and CD40. One promising approach involves the use of monoclonal antibodies that specifically target these receptors, thereby blocking their signaling pathways and potentially restoring effective immune responses against the tumor. For example, anti-CD30 therapies have shown efficacy in reducing tumor burden and improving patient outcomes by directly targeting HRS cells and altering the TME [21]. Additionally, combining these targeted therapies with immune checkpoint inhibitors (ICIs) may enhance the overall anti-tumor response by restoring the function of exhausted T cells and promoting a more favorable immune environment.

In conclusion, the role of the TME in classic HL is multifaceted, with TNF family receptors such as CD30 and CD40 serving as critical mediators of the interactions between HRS cells and the surrounding non-malignant cells. These interactions not only promote tumor survival but also facilitate immune evasion, making them attractive targets for therapeutic intervention. Continued research into the mechanisms by which HRS cells manipulate the TME will be essential for the development of novel therapies aimed at improving outcomes for patients with cHL.

## 4. Pro-Inflammatory and Anti-Inflammatory Roles of TNF Family Members

Members of the TNF family are integral to the pathophysiology of cHL, exerting both pro-inflammatory and anti-inflammatory effects that shape the tumor microenvironment. TNFR family members, such as CD30 and CD40, are overexpressed on HRS cells, mediating critical survival signals via pathways like NF-κB and JAK-STAT, which drive cell proliferation, inhibit apoptosis, and promote immune evasion. TNF-α, along with cytokines such as IL-6 and IL-1β, contributes to a pro-inflammatory microenvironment by recruiting eosinophils, mast cells, and Th2-polarized T cells, which enhance tumor growth through angiogenesis, stromal remodeling, and suppression of cytotoxic immune responses. These cytokines also induce chemokines such as CCL5, CCL17, and CCL22, which further attract immune-suppressive regulatory T cells (Tregs), reinforcing an immune-evasive niche [5,21,32,33].

Concurrently, anti-inflammatory mediators like IL-10 and TGF-β play key roles in dampening effective anti-tumor immune responses. IL-10, secreted by both HRS cells and tumor-associated macrophages (TAMs), suppresses Th1-mediated immunity and cytotoxic T cell activation, while TGF-β inhibits T cell proliferation and promotes fibrosis, which is a hallmark of the nodular sclerosis subtype of cHL. The immunosuppressive effects of these cytokines are compounded by the polarization of macrophages towards an M2 phenotype, characterized by secretion of anti-inflammatory factors that foster immune escape and further support HRS cell survival [5,21,34].

Interestingly, this cytokine imbalance is influenced by Epstein–Barr virus (EBV) infection in approximately 40% of cHL cases. EBV contributes to the dysregulated cytokine milieu by inducing the expression of viral proteins that mimic B cell survival signals, driving IL-10 production, and enhancing the recruitment of immune-suppressive cells. This dual role of TNF family members highlights their importance as both promoters of tumor growth and immune escape mechanisms. Consequently, therapeutic strategies targeting these pathways, such as inhibitors of CD30 and immune checkpoint blockade with PD-1/PD-L1 inhibitors, are being explored to disrupt these interactions, offering promising avenues to improve outcomes for cHL patients [21,33,34].

## 5. Immune Evasion and Tumor Survival

cHL utilizes a range of immune evasion mechanisms that are pivotal for tumor survival and progression. Central to these strategies are the HRS cells, which, despite being a minority in the tumor, heavily influence the surrounding immune microenvironment. HRS cells secrete chemokines such as CCL5, CCL17, and CCL20 to recruit Th2 and Tregs, fostering a supportive environment that inhibits cytotoxic T cell and natural killer (NK) cell activity [24,35,36,37]. Furthermore, HRS cells often downregulate or mutate antigen-presenting machinery, including major histocompatibility complex (MHC) class I and II molecules, impairing their recognition by immune effector cells. This is compounded by the expression of HLA-G, which protects HRS cells from NK cell-mediated lysis, and genetic alterations such as β2-microglobulin mutations that disrupt MHC-I assembly [36,37,38,39].

Additionally, HRS cells leverage immune checkpoint molecules like PD-L1 and PD-L2 to deactivate PD-1-expressing T cells in the microenvironment, thus suppressing anti-tumor immunity. Amplification of the 9p24.1 locus, which includes the PD-L1, PD-L2, and JAK2 genes, further upregulates these pathways, strengthening immune evasion [39]. HRS cells also modulate their interactions with CD4^+^ T cells by expressing surface molecules such as CD80 and CD40, which engage with immune cells to provide survival signals while simultaneously shielding the tumor from effective immune attack [35]. The loss of co-stimulatory molecules like CD58 and the epigenetic silencing of CIITA, a key regulator of MHC-II, further impair immune recognition [36,39].

The tumor microenvironment in cHL is also characterized by a significant infiltration of immunosuppressive cells, including macrophages, which are often polarized toward a tumor-supportive phenotype. HRS cells induce these macrophages to express PD-L1, creating localized immunosuppressive niches. This interplay not only protects the tumor from immune-mediated destruction but also promotes its growth and survival [39]. The complexity of these immune evasion strategies underscores the potential for targeted therapies, such as immune checkpoint inhibitors, to disrupt the tumor’s immune shield and restore anti-tumor immunity [35,36,38,39].

## 6. Checkpoint Blockade and Advanced Immunotherapies

The blockade of the PD-1/PD-L1 pathways in HRS cells represents a significant advancement in the therapeutic landscape of cHL [40,41,42,43,44]. This shift is primarily attributed to the unique TME associated with cHL, which is rich in inflammatory and immune cells that facilitate tumor immune evasion. The introduction of checkpoint inhibitors, such as nivolumab and pembrolizumab, has revolutionized treatment options for patients with relapsed or refractory cHL, showcasing the potential of harnessing the immune system to combat malignancies [23,44].

The PD-1/PD-L1 axis plays a crucial role in immune evasion by HRS cells, which express PD-L1 that interacts with PD-1 on T cells, leading to T cell exhaustion and a diminished immune response against the tumor. By blocking this interaction, checkpoint inhibitors can reactivate T cells, enhancing the immune response against HRS cells. Clinical studies have shown that the presence of PD-L1 in the TME is a key factor in the effectiveness of these therapies, correlating with the level of immune suppression and tumor progression in cHL [23,45].

Nivolumab and pembrolizumab have shown remarkable success in clinical trials, significantly improving outcomes for patients with relapsed or refractory cHL. For instance, nivolumab has been associated with an overall response rate of approximately 65% in patients who have undergone multiple lines of prior therapy, with some achieving complete remission. Similarly, pembrolizumab has demonstrated comparable efficacy, further solidifying the role of PD-1 blockade in the treatment paradigm of cHL. These results underscore the importance of the PD-1/PD-L1 pathway as a therapeutic target in a disease marked by an immunosuppressive microenvironment. The therapeutic shift toward checkpoint inhibitors is supported by an evolving understanding of the TME in cHL. In cHL, the TME includes various immune cells, such as T cells, B cells, and macrophages, that can influence the effectiveness of PD-1/PD-L1 blockade [23,28,46].

The identification of biomarkers predicting response to checkpoint inhibitors is a significant area of ongoing research, as understanding the molecular characteristics of HRS cells and the TME may enable stratification of patients most likely to benefit from PD-1/PD-L1 blockade. High levels of PD-L1 expression on HRS cells, for example, have been associated with improved responses to nivolumab and pembrolizumab, suggesting that patient selection based on TME characteristics could enhance treatment outcomes.

In conclusion, the blockade of the PD-1/PD-L1 pathways in HRS cells represents a transformative advancement in the management of cHL. The introduction of checkpoint inhibitors like nivolumab and pembrolizumab has not only improved response rates in relapsed or refractory cases but has also reshaped the understanding of the TME’s role in cancer immunotherapy [23,46].

## 7. Resistance to Chemotherapy and Immunotherapy in HL

HL demonstrates resistance to chemotherapy and immunotherapy through complex mechanisms involving the TME and immune evasion. Resistance to chemotherapy arises as HRS cells exploit protective signals within the TME. Soluble factors such as cytokines (e.g., IL-10) and chemokines (e.g., CCL5, CCL17) shield tumor cells from chemotherapy-induced cytotoxicity by reducing apoptosis and enhancing survival pathways. Direct interactions with immune-suppressive cells, including regulatory Tregs and M2-polarized macrophages, further protect HRS cells and promote drug resistance, Additionally, genetic factors such as the multidrug resistance 1 (MDR1) C3435T polymorphism have been implicated in HL susceptibility, although its role in mediating chemotherapy response is not well-established. Drug resistance is also driven by mechanisms like increased drug efflux through P-glycoprotein overexpression, underscoring the multifaceted nature of treatment resistance [36,47,48,49,50].

In immunotherapy, resistance is driven by overexpression of PD-L1 on HRS cells, mediated by 9p24.1 amplification and EBV-induced signaling pathways, which suppress T cell activity via the PD-1/PD-L1 axis. Furthermore, downregulation of MHC class I and II molecules, as well as mutations in components of the antigen presentation machinery, impair tumor recognition by cytotoxic T cells. The immune-suppressive environment, supported by myeloid-derived suppressor cells (MDSCs) and cytokines such as IL-10, further limits the efficacy of immune checkpoint inhibitors, sustaining tumor survival [48,49,51]. Resistance to BV in HL arises primarily from mechanisms such as resistance to Monomethyl Auristatin E (MMAE), the cytotoxic component, and overexpression of the MDR1 gene, which encodes P-glycoprotein, actively pumping MMAE out of cells. Unlike other cancers, CD30 expression is typically retained, suggesting resistance is mediated through intracellular drug handling rather than target loss. Additional factors contributing to resistance include the TME, which promotes resistance through metabolic reprogramming, extracellular matrix remodeling, immune suppression, and angiogenesis. Epigenetic modifications also play a role by altering the expression of key regulatory genes. Furthermore, CD30 ectodomain shedding mediated by a disintegrin and metalloproteinase (ADAM) proteins reduces effective drug docking, and defective linker-payload processing limits MMAE delivery. Strategies to address resistance include modifying payload–linker combinations, utilizing nanoparticle-based targeting, combining BV with immune checkpoint inhibitors, and exploring Clustered Regularly Interspaced Short Palindromic Repeats and CRISPR-associated protein 9 (CRISPR-Cas9) to counteract MDR1 overexpression or CD30 shedding [52,53].

## 8. The Potential of TNF Family Members as Biomarkers for HL Prognosis or Treatment Response

The TNF family and its related pathways offer potential as biomarkers in assessing HL prognosis and treatment responses. Research underscores that members of the TNF family, such as TNF-α, TRAIL, BAFF, and APRIL, and their receptors (e.g., CD30, CD40, BCMA, TACI) [1,2,3,4,5], are intricately involved in HL pathogenesis, primarily through their roles in the tumor microenvironment and their interactions with HRS cells. These interactions regulate cell survival, apoptosis, and immune evasion [33,54,55,56,57].

Elevated serum levels of TNF-α have been observed in HL patients and correlate with disease severity and poorer outcomes in some studies [58,59]. However, the role of soluble TNF receptors, though indicative of disease activity, remains complex, as these molecules may originate from both tumor and immune cells. TRAIL, another TNF superfamily member, shows potential for inducing apoptosis in HL cells when anti-apoptotic pathways are inhibited, emphasizing its utility in therapeutic strategies. Similarly, BAFF and APRIL, which support the survival of B cells, including malignant HRS cells, have been associated with lower failure-free survival in HL, highlighting their prognostic significance [33,57].

The expression of TNF-related receptors such as CD30 and CD40 on HRS cells is central to their malignant behavior. These receptors mediate interactions with immune cells, fostering an immunosuppressive microenvironment and enhancing tumor progression. CD30, in particular, has emerged as a therapeutic target, with drugs like brentuximab vedotin demonstrating efficacy in refractory HL cases. Beyond therapeutic targeting, the overexpression of TNF ligands and receptors within the tumor microenvironment further supports their utility as biomarkers [54,56].

In summary, TNF family members hold promise as biomarkers for predicting HL outcomes and guiding treatment decisions. Their dual roles in the TME—facilitating tumor growth and serving as therapeutic targets—underscore the importance of integrating these pathways into biomarker panels for a more personalized approach to HL management.

## 9. New Therapeutic Opportunities

ICIs, such as nivolumab and pembrolizumab, have demonstrated high response rates in patients with cHL, particularly in those who have not responded to conventional therapies. For instance, nivolumab has been shown to achieve an objective response rate of approximately 70% in relapsed or refractory cHL, with many patients experiencing durable remissions [60,61,62].

The mechanism of action of these inhibitors involves blocking the interaction between PD-1 on T cells and PD-L1 on tumor cells, thereby reinvigorating the immune response against the malignancy. This is particularly relevant in cHL, where HRS cells often express PD-L1, facilitating immune evasion [60]. The integration of ICIs with chemotherapy regimens has been a focal point of research. Studies have indicated that combining nivolumab with standard chemotherapy regimens, such as ABVD (doxorubicin, bleomycin, vinblastine, and dacarbazine), can improve outcomes for patients with advanced-stage cHL [63].

The rationale behind this combination is that while chemotherapy targets rapidly dividing cells, ICIs can enhance the immune system’s ability to recognize and attack residual cancer cells post-chemotherapy. This synergistic approach aims to reduce the risk of relapse and improve long-term survival rates [63]. However, the use of ICIs is associated with a unique spectrum of immune-related adverse events (irAEs), which can complicate treatment. These irAEs can affect various organ systems, including the skin, gastrointestinal tract, liver, and endocrine organs, and may require prompt recognition and management [63].

The management of these toxicities is critical, as they can lead to treatment discontinuation or dose modification, potentially impacting overall efficacy. Guidelines suggest a multidisciplinary approach to managing irAEs, which may include the use of corticosteroids or other immunosuppressive agents to mitigate severe reactions while maintaining the effectiveness of the ICI therapy [63]. In addition to managing irAEs, ongoing research is focused on identifying biomarkers that can predict responses to ICI therapy. The expression of PD-L1 on tumor cells has been associated with better responses to PD-1 inhibitors, although this relationship is not entirely consistent across all patients [62,64].

Understanding the TME and the presence of immune suppressive cells, such as regulatory T cells and myeloid-derived suppressor cells, may also provide insights into resistance mechanisms and help tailor combination therapies for individual patients [62].

The potential for combining ICIs with other therapeutic modalities, such as targeted therapies and novel agents like brentuximab vedotin, is also being explored [65]. Brentuximab vedotin, an antibody–drug conjugate targeting CD30, has shown efficacy in patients with relapsed or refractory cHL and is being studied in combination with ICIs to enhance therapeutic outcomes [64,66]. This combination approach aims to leverage the strengths of both immunotherapy and targeted therapy, potentially leading to improved response rates and reduced relapse rates. Moreover, the role of radiotherapy in conjunction with ICIs is being investigated. While traditional radiotherapy has been a cornerstone in the treatment of localized cHL, its integration with ICIs could enhance the immune response by inducing immunogenic cell death and promoting the release of tumor antigens [63].

This strategy may help to overcome some of the limitations associated with conventional therapies and improve overall treatment efficacy. In summary, the landscape of cHL treatment is rapidly evolving with the incorporation of ICIs into standard therapeutic regimens. The combination of ICIs with chemotherapy, targeted therapies, and radiotherapy presents a promising avenue for enhancing treatment efficacy while addressing the management of immune-related adverse events [67,68,69,70] (Table 2). Ongoing research into biomarkers and resistance mechanisms will be crucial in optimizing these combination strategies and personalizing treatment for patients with cHL.

## 10. Comparative Analysis of Pathobiological Features and Therapeutic Opportunities Linked to TNF Family Member Expression in cHL Across Time Periods

In the early stages of research on TNF family members, the focus was predominantly on their direct signaling capabilities. For instance, the role of TNF-alpha in inducing apoptosis and its involvement in inflammatory responses were well documented, leading to the development of therapeutic agents targeting TNF receptors directly. However, as our understanding of the immune system evolved, it became clear that the interactions between TNF family members and various immune cells within the TME are far more complex than previously thought.

The cHL microenvironment exemplifies this complexity, where HRS cells interact with a diverse array of immune cells, including T and B cells, macrophages, and eosinophils, creating a dynamic milieu that influences tumor growth and immune evasion [21].

The constitutive activation of NF-κB and alterations in the JAK-STAT signaling pathway in HRS cells contribute to the overexpression of TNF receptor family members, such as CD30 and CD40, highlighting the importance of these interactions in the context of tumor biology [21].

As the understanding of the TME deepened, researchers began to appreciate how the immune context can modulate the effects of TNF family signaling. For example, the presence of immune checkpoint molecules such as PD-1 and PD-L1 in the TME can significantly alter the response to TNF family signaling, leading to immune evasion and tumor progression. This realization has prompted a shift in treatment strategies from merely targeting TNF family receptors to leveraging immune checkpoint inhibitors that can enhance anti-tumor immunity.

The lessons learned from cHL, where the TME is rich in inflammatory cells, have been particularly instructive in this regard. Studies have shown that checkpoint blockade therapies can be more effective in tumors with a robust immune infiltrate, suggesting that the TME’s composition is a critical determinant of therapeutic outcomes [23].

The evolution of treatment strategies reflects this growing understanding of the TME’s role in modulating TNF family signaling. Early therapeutic approaches focused on directly inhibiting TNF-alpha or its receptors, which provided some clinical benefit but often fell short in terms of long-term efficacy. The emergence of immune checkpoint inhibitors, such as those targeting PD-1 and PD-L1, represents a paradigm shift in how we approach cancer treatment. These agents work by reinvigorating exhausted T cells within the TME, thereby enhancing the immune response against tumors that may have previously escaped immune surveillance due to the immunosuppressive effects of the TME [23]. This shift underscores the importance of considering the TME not just as a passive backdrop for tumor growth, but as an active participant in the regulation of immune responses. Furthermore, the interplay between TNF family members and the immune microenvironment has implications for the development of combination therapies. By understanding how TNF signaling interacts with immune checkpoints, researchers can design more effective treatment regimens that simultaneously target multiple pathways [56] (Table 3).

## 11. Conclusions

The study of TNF family involvement in HL spans nearly four decades, with initial findings in 1985 and 1995 identifying CD30 and CD40, respectively, on HRS cells and recognizing that these TNF family members play a key role in HRS cell growth and interaction with the immune system [25]. Subsequent research furthered our understanding of HL’s TME and how it supports immune escape, revealing that cytokine networks around malignant cells significantly contribute to tumor proliferation and immune suppression [21]. Recent studies have shown that targeting the PD-1/PD-L1 pathway in the TME enhances immunotherapy, marking a promising advancement in immune-targeting therapies [23].

Prospectively, combining TNF inhibitors with checkpoint blockade therapies may enhance the overall anti-tumor response by addressing both direct tumor signaling and the immune evasion mechanisms employed by tumors [71,72,73] (Figure 1). This integrated approach, which has already demonstrated effectiveness in solid tumors, is particularly relevant in the context of cHL, where the unique characteristics of the TME can be exploited to improve therapeutic outcomes [23]. Further insights into microenvironment biology and clinical trials testing predictive biomarkers and novel compounds (Table 4) will likely lead to additional improvements in outcomes.

## Figures and Tables

**Figure 1 cancers-16-04070-f001:**
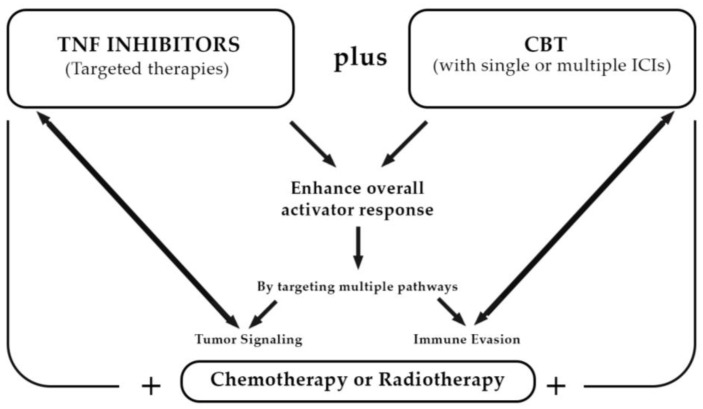
New strategy using integrated approach and combination therapies in classic Hodgkin lymphoma.

**Table 1 cancers-16-04070-t001:** Key members of the TNF ligand and receptor families [1,3].

Ligand	Ligand Symbol	Receptor(s)	Receptor Symbol(s)
tumor necrosis factor-alpha	TNF-α (TNFSF2) *	TNFR1, TNFR2	TNFRSF1A, TNFRSF1B
lymphotoxin-alpha	LT-α (TNFSF1)	TNFR1	TNFRSF1A
lymphotoxin-beta	LT-β (TNFSF3)	LTβR	TNFRSF3
OX40 ligand	OX40L (TNFSF4)	OX40	TNFRSF4
CD40 ligand *	CD40L (TNFSF5)	CD40 *	TNFRSF5
fas ligand	FasL (TNFSF6)	Fas	TNFRSF6
CD27 ligand	CD27L (TNFSF7)	CD27	TNFRSF7
CD30 ligand *	CD30L (TNFSF8)	CD30 *	TNFRSF8
4-1BB ligand	4-1BBL (TNFSF9)	4-1BB	TNFRSF9
TNF-related apoptosis-inducing ligand	TRAIL (TNFSF10)	TRAIL-R1, TRAIL-R2, TRAIL-R3, TRAIL-R4	TNFRSF10A, TNFRSF10B, TNFRSF10C, TNFRSF10D
receptor activator of nuclear factor kappa-B ligand	RANKL (TNFSF11)	RANK	TNFRSF11A
TNF-related weak inducer of apoptosis	TWEAK (TNFSF12)	Fn14	TNFRSF12A
a proliferation-inducing ligand *	APRIL (TNFSF13) *	BCMA, TACI	TNFRSF13A, TNFRSF13B
B-cell-activating factor *	BAFF (TNFSF13B)	BAFF-R *, BCMA, TACI	TNFRSF13C, TNFRSF13A, TNFRSF13B
LIGHT	LIGHT (TNFSF14)	LTβR, HVEM, DcR3	TNFRSF3, TNFRSF14, TNFRSF6B
TL1A	TL1A (TNFSF15)	DR3, DcR3	TNFRSF25, TNFRSF6B
glucocorticoid-induced TNFR ligand	GITRL (TNFSF18)	GITR	TNFRSF18
ectodysplasin A1	EDA-A1 (TNFSF14)	EDAR	TNFRSF27
ectodysplasin A2	EDA-A2	XEDAR	TNFRSF27

This table outlines key members of the TNF ligand and receptor families, which play essential roles in immune regulation, cell survival, and apoptosis. Receptors or families marked with an asterisk (*) are specifically discussed in this study, highlighting their roles in classic Hodgkin lymphoma and their therapeutic potential.

**Table 2 cancers-16-04070-t002:** Selected studies using a combination of immune checkpoint inhibitors (ICIs) with chemotherapy, targeted therapy, and radiotherapy.

Combination Therapy	Possible Benefits	References
ICIs with chemotherapy (e.g., Nivolumab + AVD)	N+AVD showed longer progression-free survival and a more favorable side-effect profile compared to BV+AVD in adolescents and adults with advanced-stage (III or IV) classic Hodgkin lymphoma	https://doi.org/10.1056/NEJMoa2405888 Herrera et al., 2024 [69]
ICIs with targeted therapy (Brentuximab Vedotin)	BV+Nivo as a first salvage therapy demonstrated high response rates and durable progression-free survival with manageable side effects in relapsed/refractory classic Hodgkin lymphoma, especially benefiting those proceeding directly to autologous stem cell transplant	https://doi.org/10.1182/blood.2020009178 Advani et al., 2021 [67]
ICIs (nivolumab, pembrolizumab) with radiotherapy	ICI-RT showed high response rates and durable complete remission in relapsed/refractory Hodgkin lymphoma, serving as an effective bridge to stem cell transplant with no major safety issues	https://doi.org/10.4081/hr.2021.9080 Lucchini et al., 2021 [70]

**Abbreviations**: AVD: adriamycin (Doxorubicin), vinblastine, dacarbazine, BV: brentuximab vedotin, ICIs: immune checkpoint inhibitors, Nivo: nivolumab, RT: radiotherapy.

**Table 3 cancers-16-04070-t003:** Molecular targets of classic Hodgkin lymphoma and therapeutic agents.

Microenvironmental Cell Markers	Therapeutic Agent
PD-1	Checkpoint inhibitors (Nivolumab, Pembrolizumab)
PD-L1	Checkpoint inhibitors
**Deregulated pathways**	
NFκB	NFκB inhibitors (Bortezomib)
AKT/MAPK	MAPK inhibitors (Sorafenib)
	HDAC inhibitors
PI3K/AKT	Idelalisib
mTOR	mTOR inhibitor (Everolimus)
**HRS markers**	
CD30	Monoclonal antibodies anti-CD30 (Brentuximab Vedotin)
CD40	Monoclonal antibodies anti-CD40
MUM1/IRF4	Immunomodulator (Lenalidomide)
Notch1	Notch inhibitor
CD20	Monoclonal antibodies anti-CD20 (Rituximab)
CD80	Monoclonal antibodies anti-CD80 (Galiximab)
HDACs	HDAC inhibitors
EBV	Patient-derived cytotoxic T cells

**Abbreviations**: AKT/MAPK: AKT/Mitogen-activated protein kinase, CD20: cluster of differentiation 20, CD30: cluster of differentiation 30, CD40: cluster of differentiation 40, CD80: cluster of differentiation 80, EBV: Epstein–Barr Virus, HDACs: histone deacetylases, mTOR: mammalian target of rapamycin, MUM1/IRF4: multiple myeloma oncogene 1/Interferon regulatory factor 4, NF-κB: nuclear factor kappa B, Notch1: notch homolog 1, PD-1: programmed cell death protein 1, PD-L1: programmed death-ligand 1, PI3K/AKT: phosphoinositide 3-kinase/AKT.

**Table 4 cancers-16-04070-t004:** Selection of ongoing phase I/II trials evaluating novel therapeutic combination in classic Hodgkin lymphoma (cHL).

Target	Agent/Intervention	Phase	Clinical Setting	Trial Identifier
CD30	Autologous CAR.CD30 EBV specific-CTLs	1	Relapsed CD30+ Hodgkin lymphoma and CD30+ non-Hodgkin lymphoma	NCT01192464
CD30	CD30 CAR T cell	1/2	Post-AutoHSCT for poor-risk Hodgkin lymphoma	NCT06617286
CD30	LCAR-HL30 cells	1	Relapsed/refractory Hodgkin lymphoma and anaplastic large cell lymphoma	NCT06494371
CD30	Modified BV-AVD-R regimen	2	Children with intermediate/high-risk cHL	NCT06201507
CD30	ATLCAR.CD30.CCR4 cells	1/2	Relapsed/refractory CD30+ Hodgkin lymphoma	NCT06090864
CD30	GEN3017	1/2	Relapsed/refractory CD30+ Hodgkin and non-Hodgkin lymphoma	NCT06018129
CD30	AFM13 + AB-101	2	Recurrent/refractory Hodgkin lymphoma and CD30+ peripheral T cell lymphoma	NCT05883449
CD30	Brentuximab Vedotin + CHP	2	Newly diagnosed CD30+ peripheral T cell lymphoma (PTCL)	NCT05673785
CD30	Brentuximab Vedotin + AVD	Observational	Pediatric patients with newly diagnosed CD30+ Hodgkin lymphoma	NCT05481437
CD30	CD30-directed CAR-T cells in combination with Nivolumab	1	R\R cHL post-frontline therapy failure	NCT05352828
CD30	Anti-CD30 CAR-T Cell Injection	1	R\R CD30+ lymphoma	NCT05208853
CD30	BV	Observational	R\R CD30+ lymphoma	NCT04998331
CD30	HSP-CAR30 T cells	1/2	R\R Hodgkin and T cell lymphoma	NCT04653649
LAG-3, PD-1, CD47	AK129 with or without AK117	1/2	R\R classic Hodgkin lymphoma, PD-1/L1 inhibitor treatment failure	NCT06642792
PD-1	Pembrolizumab	2	First-line treatment, advanced-stage cHL	NCT06045195
PD-1	Tislelizumab	2	Frontline, de novo Hodgkin lymphoma unsuitable for standard chemotherapy	NCT05977673
PD-1	Zimberelimab + AVD	2	First-line, early-stage Hodgkin lymphoma	NCT05900765
PD-1	Prolgolimab ± Bendamustine	2	Second-line, relapsed R\R cHL	NCT05757466
CD30 + PD-1	Autologous CD30.CAR-T + Nivolumab	1	R\R cHL after frontline therapy	NCT05352828
PD-1	PD-1 inhibitor + Decitabine + ASCT	2	Second-line for R\R cHL	NCT05137886
PD-1	PD-1 inhibitor ± GVD regimen	2	R\R cHL	NCT04624984
JAK	Ruxolitinib	2	R\R cHL	NCT02164500
JAK	INCB047986	1	Advanced malignancies, including Hodgkin and non-Hodgkin lymphoma	NCT01929941
CD30	CD30 CAR T cell	1/2	Poor-risk cHL post-AutoHSCT	NCT06617286
CD30	HSP-CAR30	1/2	R\R Hodgkin and T cell lymphoma	NCT04653649
PD-1	PD-1 inhibitor	1	R\R Hodgkin lymphoma	NCT04134325
BTK	Ibrutinib	2	R\R cHL	NCT02824029
CD30	CD30.CAR-EBVST cells	1	R\R CD30-positive lymphomas	NCT04288726
CD30, PD-1	Chidamide + Decitabine + Anti-PD-1 antibody and Brentuximab Vedotin + Bendamustine + Anti-PD-1 antibody	2	R/R cHL, transplant-ineligible, or refused a transplant	NCT06563778
PD-1	Pembrolizumab coformulated with Hyaluronidase	2	R/R cHL or R/R PMBCL	NCT06504394
PD-1	Chidamide + Decitabine + Anti-PD-1 antibody (experimental), Brentuximab Vedotin + Anti-PD-1 antibody (comparator)	2	R/R cHL for ineligible patients or those refusing a transplant	NCT06393361
PD-1	Azacitidine + PD-1 therapy	2	Relapsed/refractory classic Hodgkin lymphoma	NCT06190067

**Abbreviations**: AK117: experimental drug targeting CD47, AK129: experimental drug targeting PD-1, ASCT: autologous stem cell transplant, AutoHSCT: autologous hematopoietic stem cell transplant, AVD: Adriamycin (doxorubicin), Vinblastine, and Dacarbazine, BTK: Bruton’s Tyrosine Kinase, CAR-T: Chimeric Antigen Receptor T cells, cHL: classic Hodgkin lymphoma, EBVST cells: Epstein–Barr virus-specific T cells, GVD: Gemcitabine, Vinorelbine, and Doxorubicin, HSP-CAR30 T cells: Heat Shock Protein–Chimeric Antigen Receptor targeting CD30 T cells, INCB047986: experimental JAK inhibitor, JAK: Janus Kinase, LAG-3: Lymphocyte Activation Gene-3, PD-1: Programmed Death-1, PMBCL: primary mediastinal B cell lymphoma, R/R: relapsed/refractory.

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
