# Peer review of "Pathobiological Features and Therapeutic Opportunities Linked to TNF Family Member Expression in Classic Hodgkin Lymphoma"

_cancers, 2024, doi:10.3390/cancers16234070_

Round 1

Reviewer 1 Report

Comments and Suggestions for Authors

Thank you for the opportunity to review the manuscript of Alibraim MN et al, which is a review of the therapeutic approaches of cHL written by a reputed group studying cHL for years. The manuscript is well and clearly written, and it offers interesting information for the reader. I only have minor questions to be discussed.

The authors emphasize the role of CD40 as well as CD30 in the pathogenesis of cHL, and suggest the interest of CD40 as a potential therapeutic target. However, CD40 is not a common marker included in the diagnostic workup of cHL. So that, do the authors recommend to include CD40 in the routine diagnostic panel for cHL?

Would they recommend other proteins for diagnostic/therapeutic purposes?

In addition, based on the experience of the group, could be CD40 useful to distinguish cHL from other entities and other lymphomas with overlapping features with cHL?

Author Response

Thank you for the opportunity to review the manuscript of Alibraim MN et al, which is a review of the therapeutic approaches of cHL written by a reputed group studying cHL for years. The manuscript is well and clearly written, and it offers interesting information for the reader. I only have minor questions to be discussed.

Thank you

The authors emphasize the role of CD40 as well as CD30 in the pathogenesis of cHL, and suggest the interest of CD40 as a potential therapeutic target. However, CD40 is not a common marker included in the diagnostic workup of cHL. So that, do the authors recommend to include CD40 in the routine diagnostic panel for cHL?

On line 113 of the revised manuscript we specify that CD40 is expressed in 100% of cases, so demonstrating its utility in recognizing HRS cells, without emphasizing its diagnostic utility because CD40 is largely expressed in B-cell lymphomas.

Would they recommend other proteins for diagnostic/therapeutic purposes?

Presently, we do not recommend other proteins for diagnostic/therapeutic purpose. See also Table 3 regarding HRS cell markers.

In addition, based on the experience of the group, could be CD40 useful to distinguish cHL from other entities and other lymphomas with overlapping features with cHL?

CD40 is not useful to distinguish cHL from other entities and other lymphomas with overlapping features with cHL, because it is largely expressed in B-cell non-Hodgkin lymphomas.

Reviewer 2 Report

Comments and Suggestions for Authors

cancers-3328768

Type of manuscript: Review

Title: Pathobiologic features and therapeutic opportunities linked to TNF family members expression in classic Hodgkin lymphoma

Authors: Mohamed Nazem Alibrahim *, Annunziata Gloghini, Antonino Carbone *

This review article is about the role of the TNF family in the treatment of Hodgkin lymphoma. Considering the importance of these two key terms, I believe it is a highly necessary review article, but as pointed out below, it requires many revisions.

[Major concerns]

1.   Generally, there are two formats for writing a review paper: narrative review and systematic review. In the case of this review paper, it would be helpful if the author specifies one of these formats and describes the corresponding writing method, literature search, and other related aspects in the introduction. This would make it much easier for readers to follow.

2.   Numbers of references: I was also quite surprised to see that only 36 references were cited. In all my experience reviewing hundreds of research and review papers, this is the first time I have encountered a review article with such a limited number of references. A search (as of November 12, 2024) using the keywords "Hodgkin lymphoma" and "TNF family" on PubMed yields 69 articles, and when adding the keyword "Review," 19 articles are found. In other words, I believe the review article is lacking depth in the research with only 36 references. Typically, research articles cite about 30 to 50 references, while review articles often cite around 100 references.

3.   There are several unresolved debates regarding the functions and roles of TNF family members in the treatment of Hodgkin lymphoma (HL). These include: 1) the pro-inflammatory and anti-inflammatory roles of TNF family members, 2) immune evasion and tumour survival, 3) the tumour microenvironment, 4) resistance to chemotherapy and immunotherapy in HL, and 5) the potential of TNF family members as biomarkers for HL prognosis or treatment response. In this paper, the focus was primarily on the tumour microenvironment, while the other issues were somewhat neglected. In my opinion, it would be beneficial to expand on the issues that were not addressed, perhaps by increasing the number of references, to provide a more comprehensive discussion.

4.   Abbreviations: The use of abbreviations when writing a paper has many advantages besides simplicity of expression. To use an abbreviation, first write the abbreviation in parentheses after the full name, and then use the abbreviation from Introduction to the final Conclusion. Abbreviations should only be used if they are repeatedly used and if they are not used again, only the full name should be used.

5.   In cases where abbreviations are used within figures or tables, please list these abbreviations along with their corresponding full names in the figure legends or at the bottom of corresponding tables. If there are two or more abbreviations, arrange them in alphabetical order of the abbreviations. In this paper, the first letter of the words corresponding to acronyms is capitalized, so selectively write them all in lowercase.

6.   English: There are many instances in the paper where non-proper nouns are capitalized within sentences. Please find all such cases and change them to lowercase. Examples: Tumour Necrosis Factor at Lines 16 and 38; etc.

7.   The order of CD30 and CD40 notation: Unless there is a specific reason, please list them in numerical order (using Arabic numerals) in both the text and figures.

8.   Figures 1 and 2: In a review paper, unpublished experimental data are not included. Figures 1 and 2 in this paper fall into this category. Given the nature of this review, it is not necessary to include these two immunohistochemical microphotographs. Additionally, since it is well-known that CD30 and CD40 are expressed in HL, and the immunological characteristics of Reed-Sternberg cells in HL are widely understood, it would be sufficient to simply describe these points in the text.

9.   Table 2 and Figure 3: The fonts used in Table 2 and Figure 3 are different from the surrounding fonts or appear awkward, so please correct them.

[Minor concerns]

1.    Line 23: Define PD-1 and PD-L1 in the Abstract.

2.    Line 52: Define TNFR1.

3.    Line 71: NF-kappaB should be written as NF-κB. There are many same typos in the paper.

4.    References: Adjust the citation style of the references to conform to the Cancers guidelines, and ensure that any missing page numbers are accurately included. Examples: 22, 25, 28, etc.

Overall, the manuscript can be considered to publication after major revision as indicated above.

Comments on the Quality of English Language

cancers-3328768

Type of manuscript: Review

Title: Pathobiologic features and therapeutic opportunities linked to TNF family members expression in classic Hodgkin lymphoma

Authors: Mohamed Nazem Alibrahim *, Annunziata Gloghini, Antonino Carbone *

This review article is about the role of the TNF family in the treatment of Hodgkin lymphoma. Considering the importance of these two key terms, I believe it is a highly necessary review article, but as pointed out below, it requires many revisions.

[Major concerns]

1.   Generally, there are two formats for writing a review paper: narrative review and systematic review. In the case of this review paper, it would be helpful if the author specifies one of these formats and describes the corresponding writing method, literature search, and other related aspects in the introduction. This would make it much easier for readers to follow.

2.   Numbers of references: I was also quite surprised to see that only 36 references were cited. In all my experience reviewing hundreds of research and review papers, this is the first time I have encountered a review article with such a limited number of references. A search (as of November 12, 2024) using the keywords "Hodgkin lymphoma" and "TNF family" on PubMed yields 69 articles, and when adding the keyword "Review," 19 articles are found. In other words, I believe the review article is lacking depth in the research with only 36 references. Typically, research articles cite about 30 to 50 references, while review articles often cite around 100 references.

3.   There are several unresolved debates regarding the functions and roles of TNF family members in the treatment of Hodgkin lymphoma (HL). These include: 1) the pro-inflammatory and anti-inflammatory roles of TNF family members, 2) immune evasion and tumour survival, 3) the tumour microenvironment, 4) resistance to chemotherapy and immunotherapy in HL, and 5) the potential of TNF family members as biomarkers for HL prognosis or treatment response. In this paper, the focus was primarily on the tumour microenvironment, while the other issues were somewhat neglected. In my opinion, it would be beneficial to expand on the issues that were not addressed, perhaps by increasing the number of references, to provide a more comprehensive discussion.

4.   Abbreviations: The use of abbreviations when writing a paper has many advantages besides simplicity of expression. To use an abbreviation, first write the abbreviation in parentheses after the full name, and then use the abbreviation from Introduction to the final Conclusion. Abbreviations should only be used if they are repeatedly used and if they are not used again, only the full name should be used.

5.   In cases where abbreviations are used within figures or tables, please list these abbreviations along with their corresponding full names in the figure legends or at the bottom of corresponding tables. If there are two or more abbreviations, arrange them in alphabetical order of the abbreviations. In this paper, the first letter of the words corresponding to acronyms is capitalized, so selectively write them all in lowercase.

6.   English: There are many instances in the paper where non-proper nouns are capitalized within sentences. Please find all such cases and change them to lowercase. Examples: Tumour Necrosis Factor at Lines 16 and 38; etc.

7.   The order of CD30 and CD40 notation: Unless there is a specific reason, please list them in numerical order (using Arabic numerals) in both the text and figures.

8.   Figures 1 and 2: In a review paper, unpublished experimental data are not included. Figures 1 and 2 in this paper fall into this category. Given the nature of this review, it is not necessary to include these two immunohistochemical microphotographs. Additionally, since it is well-known that CD30 and CD40 are expressed in HL, and the immunological characteristics of Reed-Sternberg cells in HL are widely understood, it would be sufficient to simply describe these points in the text.

9.   Table 2 and Figure 3: The fonts used in Table 2 and Figure 3 are different from the surrounding fonts or appear awkward, so please correct them.

[Minor concerns]

1.    Line 23: Define PD-1 and PD-L1 in the Abstract.

2.    Line 52: Define TNFR1.

3.    Line 71: NF-kappaB should be written as NF-κB. There are many same typos in the paper.

4.    References: Adjust the citation style of the references to conform to the Cancers guidelines, and ensure that any missing page numbers are accurately included. Examples: 22, 25, 28, etc.

Overall, the manuscript can be considered to publication after major revision as indicated above.

Author Response

Please see the attached file "Reviewer 2"

Round 2

Reviewer 2 Report

Comments and Suggestions for Authors

 Accept in present revised form.

However, there are several typos. Technical corrections are needed at the Office.